# Shunt Overdrainage: Reappraisal of the Syndrome and Proposal for an Integrative Model

**DOI:** 10.3390/jcm10163620

**Published:** 2021-08-17

**Authors:** Bienvenido Ros, Sara Iglesias, Jorge Linares, Laura Cerro, Julia Casado, Miguel Angel Arráez

**Affiliations:** 1Pediatric Neurosurgery Division, Department of Neurosurgery, Regional University Hospital, Avda. Carlos Haya s/n, 29010 Malaga, Spain; sara.iglesias.sspa@juntadeandalucia.es; 2Department of Neurosurgery, Regional University Hospital, Avda. Carlos Haya s/n, 29010 Malaga, Spain; jorge.linares.sspa@juntadeandalucia.es (J.L.); laura.cerro.sspa@juntadeandalucia.es (L.C.); julia.casado.sspa@juntadeandalucia.es (J.C.); marraezs@uma.es (M.A.A.); 3Department of Surgery, Malaga University, 29010 Malaga, Spain

**Keywords:** antisiphon device, craniocerebral disproportion, gravitational valves, pathophysiology, shunt overdrainage, siphoning, slit ventricle syndrome

## Abstract

Although shunt overdrainage is a well-known complication in hydrocephalus management, the problem has been underestimated. Current literature suggests that the topic requires more examination. An insight into this condition is limited by a lack of universally agreed-upon diagnostic criteria, heterogeneity of published series, the multitude of different management options and misunderstanding of relationships among pathophysiological mechanisms involved. We carried out a review of the literature on clinical, radiological, intracranial pressure (ICP), pathophysiological and treatment concepts to finally propose an integrative model. Active prophylaxis and management are proposed according to this model based on determination of pathophysiological mechanisms and predisposing factors behind each individual case. As pathophysiology is progressively multifactorial, prevention of siphoning with gravitational valves or antisiphon devices is mandatory to avoid or minimize further complications. Shunt optimization or transferal and neuroendoscopy may be recommended when ventricular collapse and cerebrospinal fluid isolation appear. Cranial expansion may be useful in congenital or acquired craniocerebral disproportion and shunting the subarachnoid space in communicating venous hydrocephalus and idiopathic intracranial hypertension.

## 1. Evolution of Concepts and Current Pitfalls in Shunt Overdrainage Syndrome

Early historical cases of excessive drainage of brain fluid have been previously reported in the literature [1,2,3,4,5,6,7]. In the contemporary era of neurosurgery, Fox and coworkers were the first to report ICP monitoring findings in shunted patients. In 18 patients with normal pressure hydrocephalus (NPH), mean cerebrospinal fluid (CSF) pressure values about −220 mm H_2_O for ventriculoperitoneal shunts (VPS) and about −190 mm H_2_O for ventriculoarterial shunts (VAS) were obtained in the upright position, these findings being explained by the siphoning action of shunts. Siphoning might be responsible for annoying postural headaches and cases of subdural hematoma and higher-pressure valves and/or VAS were recommended for patients expected to be upright for much of their waking period [8]. Portnoy contributed to this “mechanistic model” by developing an “antisiphon device” (ASD) to prevent siphoning [4,9]. ICP characteristics of postural siphoning were confirmed in 1990 by Chapman using a telemetric device in patients with VPS, VAS and ventriculopleural shunts. ASD were generally effective in restoring “normal pressures” in the upright position [10]. Pudenz also investigated the hypothesis of overdrainage caused by siphoning [11]. Siphoning was not always a negative effect of shunts: Bergsneider reported on the benefit of siphoning to induce ventricular hypotension in shunt non-responsive hydrocephalic patients [12]. 

Chronic overdrainage was also soon associated with recurrent shunt obstruction secondary to small ventricles, and subtemporal craniectomies were carried out in the 1970s to prevent this complication [13,14]. In 1982 Hyde-Rowan and coworkers first defined the so-called “slit ventricle syndrome” (SVS) as the clinical picture characterized by a triad: intermittent or chronic headaches, small ventricles on CT scan or ventriculogram, and slow refill of the palpable valve mechanism. These characteristic episodes, usually lasting from 10 to 90 min, were considered to be secondary to ventricular catheter obstruction that was, therefore, episodic as well. The authors believed that subtemporal craniectomies were not successful in the long-term so prophylactic measures should be employed to prevent slit ventricles. Avoiding low-pressure valves, valve upgrading or in-line ASD implantation were recommended whenever a slit ventricle was encountered, even in asymptomatic patients [15]. Valve obstruction was only temporary in these episodes. Shunt patency was later documented in patients with headaches and small ventricles seen on isotope scan studies [16]. 

Overdrainage itself could lead to further symptomatology, which could be divided into acute and chronic manifestations [17]. Once chronic symptoms were established, algorithms including sequenced strategies (shunt revision, cranial expansion), then were necessary [18]. In the early 1990s, Abbot described three distinct categories in shunt overdrainage according to ICP monitoring: headaches from hypotension (ICP drop when sitting as a result of shunt siphoning), intermittent headache episodes (high-pressure waves were recorded) and a third group of patients that showed no relation of symptoms to pressure. ASD implantation was recommended in the first group, shunt revision or cranial expansion in the second and no treatment, shunt removal, headache clinic or medication for migraine in the third group [16]. In 1993, Rekate proposed a subclassification of SVS divided into five distinct syndromes by adding the so-called “shunt failure with small ventricles” and “intracranial hypertension with working shunt” types [19]. Baskin implemented an ICP-based algorithm for shunt removal in SVS refractory to increased shunt resistance and medical therapy: shunt withdrawal, endoscopic third ventriculostomy (ETV) or LPS shunts were employed according to ventricular dilatation and ICP measures [20]. Later on, Eide carried out calvarial expansion in patients with intracranial hypertension, considering SVS as the result of craniocerebral disproportion (CCD) with impaired intracranial compliance [21], whereas implantation of an ASD and/or higher-pressure valve was recommended in patients with abnormally reduced ICP [22]. 

Nowadays, different classifications and algorithms have been published concerning SVS [23,24] but the most accepted concepts of shunt overdrainage are related to what has become known as “overdrainage syndromes” [25] or lately “shunt related headaches” [26]. A possible equivalence between these different classifications has been proposed [27]. Rekate considered shunt overdrainage to be an entity characterized by the appearance of severe headache, i.e., that which interferes with activities of daily living, in patients with a CSF shunt valve and normal or smaller than normal ventricles. The five known different categories have now been re-named: severe intracranial hypotension or low-pressure headaches (analogous to spinal headaches), intermittent obstruction of the ventricular catheter (“slit ventricle syndrome” itself), intracranial hypertension with small ventricles and a failed shunt, intracranial hypertension with a working shunt and headaches unrelated to shunt function (“shunt-related migraine”) [26]. These concepts, which have been globally accepted, include the whole spectrum, so a variety of symptoms, clinical presentations, radiological findings and pressure readings can be expected [28]. 

Recent reports, however, have suggested that the evidence about overdrainage is not as robust as presumed and that the topic requires yet more examination [4]. An insight into the condition is limited by a lack of universally agreed-upon diagnostic criteria, heterogeneity in the clinical characteristics of patients included in studies and a multitude of non-evidence-based management options. Besides, patients can develop different causes of their headaches over time [29]. The clinical-radiological picture is more complex than the classic descriptions of headaches in patients with small ventricles. Symptoms can also appear in patients with normal or large ventricles [30] and slow valve refilling does not necessarily mean valve malfunction or overdrainage [31]. Both the “intracranial hypertension with a working shunt” and “shunt failure without ventricular enlargement” types are particularly difficult to distinguish in practice or even directly overlap. Similarities with the so-called “normal volume hydrocephalus” of Engel [32], “shunt malfunction without ventricular dilatation” of McNatt [33] and “hydrocephalic pseudotumor” of Rekate [19] have been proposed [27]. Additionally, the term “headache or non-shunt-related migraine” seems to be a catch-all definition [27]. Terms such as CCD, SVS and chronic overdrainage are unfortunately sometimes used as synonyms [34]. On the other hand, important problems also exist regarding the pathophysiology of shunt overdrainage. Indeed, this has been considered one of the issues which we do not know (but should) about hydrocephalus [35]. In the current literature, it has been widely accepted that siphoning of CSF leads to intracranial hypotension, which may progress to SVS, but a significant number of papers are observational clinical reports that do not offer a pathophysiological hypothesis. In clinical practice, more than one pathophysiological mechanism may be involved in a single case, perhaps with one predominating which should determine the main treatment strategy [27]. On the other hand, some attention needs to be shifted to the prophylaxis of overdrainage in susceptible patients [4]. Finally, surgical management of chronic overdrainage is becoming more complex, as surgeons have to deal with new adjustable gravitational valves or different models of ASD. 

The lack of standardized clinical criteria for its diagnosis explains the different rates of shunt overdrainage (from 1% to over 50% of shunted subjects) reported [27,36]. In a recent survey sent to active ASPN members regarding shunted-hydrocephalus management, most respondents considered chronic CSF overdrainage to be a rare complication of shunting (no more than 15%) in their practice, attributing chronic headaches to medical reasons (migraines, tension) with patients being frequently referred to stress management or pain clinics [37]. There may have been undiagnosed overdrainage behind a certain percentage of shunt malfunctions [38] as overdrainage predisposes to obstruction of the ventricular catheter [39]. According to Rekate, at least one-third of chronic shunted patients followed for more than 5 years will have serious chronic headaches, although the picture could be subclinical for decades [26]. Radiological overdrainage can appear in various percentages of shunted patients, ranging from less than 10% up to even 85% [27]. In our series of 166 shunted patients that we followed for a mean of 93 months, 56% developed some form of symptomatic overdrainage, which was the most usual reason for surgical revision due to valve malfunction. Antisiphon and/or gravitational devices were used to treat symptomatic overdrainage but we did not give active prophylaxis for overdrainage during that period of time [40]. However, it is not just a matter of a high rate but also of severity. Patients who developed severe overdrainage had unpredictable responses to a gradual increase in system resistance and required several surgical revisions of their shunts before they became symptom-free [40]. Furthermore, symptomatic overdrainage was associated with a worse cognitive outcome measured by the HOQ—Spanish version [41].

In an attempt to confront these limitations, we will now consider clinical, radiological, ICP, pathophysiological and treatment concepts separately, to finally propose an integrative model for the management of shunt overdrainage syndrome (SOS).

## 2. Clinical Manifestations in Shunt Overdrainage 

Overdrainage of CSF may appear in an acute manner. Postoperative extradural hematoma [42] and the development of posterior reversible encephalopathy [43] have been described, caused by intraoperative overdrainage of CSF with rapid reduction in ICP as a result of a craniotomy or CSF diversion procedure. Regarding chronic overdrainage, we should always consider a silent period of asymptomatic (radiological) overdrainage of variable duration. Some patients do not even develop symptoms during low pressure stages [44]. When symptoms appear, they usually start as a “low-pressure headache”, headache of postural characteristics or “spinal headache”, with the patient unable to tolerate sitting up. The symptom complex may also include nuchal or upper back pain, nausea-vomiting, dizziness, fatigue, irritability, gait disturbance, diplopia, seizures, lethargy, etc. [29,45]. A sinking skin/bone flap may appear in shunted patients with decompressive craniotomies [46,47]. The “orthostatic intolerance syndrome”, a rare dysautonomic condition related to fourth ventricle dilatation, may simulate a low ICP state [48,49,50]. Compressive myeloradiculopathy may occasionally occur due to ingurgitation of the venous epidural plexus together with meningeal thickening [51,52,53]. Similarities with spontaneous intracranial hypotension (SIH) have been described by some authors [54]. This entity has been recently termed overshunting-associated myelopathy (OSAM), and an association with jugular vein occlusion has been postulated [55,56]. 

Low-pressure symptoms may evolve to intermittent headaches and the latter to chronic states with developmental delay, decline in school performance and social withdrawal. Multiple shunt revisions are frequently present in clinical records. The so-called “dorsal midbrain syndrome” (conjugate vertical gaze palsy, parkinsonism, memory loss, fluctuation in the level of consciousness, hypothalamic dysfunction, hypersalivation, peripheral facial palsy and blepharospasm) has also been reported as a result of overdrainage, upward midbrain herniation and aqueduct obstruction [57]. Finally, symptoms and signs of sustained intracranial hypertension may appear (papilledema, altered consciousness, bradycardia or systemic hypertension) [29,34,58] with management of these states sometimes being a life-threatening emergency [59]. 

## 3. Radiology in Shunt Overdrainage

In the initial phase of ventricular emptying, extra-axial collections of fluid or blood may appear, hypodense, isodense or hyperdense on CT depending on the acute, subacute or chronic blood content [42,60]. Although most patients show small ventricles on imaging, moderate- or normal-sized ventricles are also found [61]. MRI studies may show widened brain sulci, parenchymal calcifications or overshunting-associated myelopathy findings in some cases [51,52,53,55,56,62,63]. The radiological compression may not have any clinical repercussions [56]. 

The final ventricular size and shape are the result of the initial size, the reconfiguration of the cerebral mantle and the presence of porencephaly. Ventricular collapse with no CSF around the ventricular catheter tip (“tip environment”) is characteristic of SVS [64]. Sometimes, a transient increase in the size of the ventricles on imaging is observed during the period of symptom exacerbation, followed by a decrease during symptom remission [65]. The presence of compartments or sustained ventricular isolation may be a consequence of SVS [66]. 

Chronic shunting in infants can lead to premature sutural fusion (suture sclerosis) and cranial vault or skull base bone changes, such as microcephaly, dolichocephaly, laminated thickening of the calvarium or sinus hyperpneumatization [67,68]. CCD may be present in the supratentorial and infratentorial compartments (acquired Chiari) [34,69,70]. Spinal canal stenosis may also be present [71,72]. Figure 1 shows some examples with the main radiological characteristics of shunt overdrainage. 

## 4. ICP Patterns in Shunt Overdrainage

The type of overdrainage cannot be determined based solely on clinical symptoms [73]. Therefore, ICP monitoring can be a useful tool [74]. Patients with low or high ICP may present with similar symptoms and radiological findings [75] and high ICP values might be present in patients with mild symptoms or even asymptomatic patients [74]. 

Early ICP monitoring reports showed that standard differential pressure VAS, VPS and ventriculopleural shunts all caused, in normal and chronically shunted patients, a rapid, “dramatic” linear fall in intraventricular pressure as the subjects were brought into the upright position. The fall started when the angle of body elevation was greater than 25 to 35° and reached pressures in the range of −15 to −35 cm H_2_O [10] or −20 to −25 cm according to other authors [19]. Recently, standardized positional testing may identify hypotensive overdrainage patterns. A low baseline ICP (slightly positive, i.e., around 5 mmHg) in a recumbent position is followed by an immediate ICP drop to slightly negative values (i.e., −5 mmHg) in the first minute after sitting up and then by a gradual ICP decrease over the next 25 min to a minimum of less than −10 mmHg [45]. In these patients, overnight ICP monitoring studies show low or negative basal pressures (Pb) in the horizontal position with a further decrease in pressure to a value lower than −10 mmHg in tilting tests. In infusion tests, Weerakkody reported Pb in the supine position < 0 mmHg, ICP at the plateau phase lower than critical pressure for specific shunt, Rcsf < 8 mmHg/ml/min, pulse amplitude baseline < 2 mmHg and RAP index < 0.6 [44]. According to Eide, the ICP pulse pressure amplitude should stay on average below 4 mmHg in normal conditions and in shunt overdrainage, SIH and shunt non-responders. Figures above 4 mmHg are found in hydrocephalic patients prior to shunting or with shunt underdrainage. When ICP is less than −15 mmHg, a slight increase in pulse pressure amplitude can be found in adult patients [76]. 

On the other hand, chronic shunted patients may develop patterns of intracranial hypertension. In early studies, mean ICP rise > 20 and peaks > 50 mmHg were encountered in patients with “intermittent” symptoms [19]. Recently, in overnight ICP monitoring (OIM) studies, ICP recordings showed irregular traces with a mean ICP > 20 mmHg, triangular shape of ICP pulse waveform and an increase in frequency and amplitude of vasogenic waves (B wave storms and recurrent plateau waves) [28,34,74]. An ICP trace can be divided into baseline intervals and wave intervals. A relative increase in the mean ICP (with ICP peaks > 25 mmHg or even > 50 mmHg), RAP index (RAP peaks > 0.6) and in the spectral parameters AMP and SLOW (>5 mmHg) have been described during the wave episodes compared to baseline [19,28,29,45]. In infusion tests, Pb is usually high but the pulse wave rarely visible; when collapsed ventricles open, pressure stabilizes at a lower level with a pulse wave clearly visible [29,44].

Two main ICP patterns have been recently proposed by Czosnyka. Overdrainage related to ICP slow vasogenic waves has been proposed as opposed to postural overdrainage related to siphoning [77]. Figure 2 and Figure 3 illustrate differences between hypotensive and hypertensive patterns is shunt overdrainage.

## 5. Pathophysiological Theories in Shunt Overdrainage

Different pathophysiological theories have been developed to describe the characteristic behavior of every single type of chronic shunt overdrainage. The condition, however, is clearly progressive and, therefore, the combination of mechanisms in a specific order and proportion occurs. The following suggested theories are analyzed individually to better understand their integration into a global theory (Figure 4).

### 5.1. Siphoning

The role of postural change is explained by the law of Stevin: PP = (ICP − IAP) + HP, where PP is the perfusion pressure, ICP the intracranial pressure, IAP the intra-abdominal pressure and HP the hydrostatic pressure. When the patient goes from decubitus to an upright position a gradient appears due to gravity between the ventricles and the peritoneal cavity that is equivalent to the weight of the column of CSF inside the system (hydrostatic pressure). This “negative hydrostatic suction force” depends on the distance between both cavities and can easily exceed the opening pressure, resulting in siphoning, ventricular emptying (leading to “collapse”) with intracranial hypotension, reduction in cerebral volume and expansion of the subarachnoid spaces or development of subdural CSF or blood effusions [36]. These complications are rare in infants because of the more elastic properties of the brain which limit the effects of brain shrinkage but conditions with white matter damage (post-hemorrhagic hydrocephalus-PHH-, aqueductal stenosis-AS-) are related with higher rates. In adults, previous subdural hematoma (SDH) or hygroma (SDHy), post-traumatic hydrocephalus (PTH) or chronic conditions with large ventricles and low brain elastance (LOVA, NPH or brain atrophy), favor siphoning-related complications [36,78,79,80]. Siphoning may also trigger other pathophysiological mechanisms. 

### 5.2. Ventricular Collapse and CSF Isolation

When overdrainage leads to ventricular collapse onto the catheter tip, an intermittent or prolonged blocking of the shunt occurs and therefore CSF becomes isolated (stuck) from the main stream of circulation (the shunt). Distortion of the cerebral structures that follows CSF drainage may result in post-shunt ventricular asymmetry leading to unilateral collapse (with contralateral isolated ventricle) as a consequence of closure of the Monro by septum displacement in frontal catheters or medial displacement of the thalamus in atrial catheters [66,81,82]. In other cases, the pressure gradient between the supratentorial and infratentorial ventricular cavities leads to upward displacement of the midline cerebellar structures into the tentorial incisura, with subsequent distortion of the aqueduct. Trapping of the fourth ventricle may occur in the presence of membranes blocking fourth ventricle outlets due to post-infective or posthemorrhagic inflammatory processes or chemical or radiation-induced ventriculitis [36]. In communicating hydrocephalus, when the whole ventricular system is collapsed, CSF may become isolated in the subarachnoid space (with raised ICP in this compartment) and a “shunt pseudotumor” develops [25,83].

Predisposing factors for ventricular collapse and CSF isolation include shunt implantation during the first months of life [36,68,83], PHH of the newborn or premature [38,60,83] or neonatal meningitis [60,83], valves functioning well for several years [68,84] and low-pressure opening valves [36,84,85]. On the other hand, this phenomenon is unlikely to occur in patients with severe brain atrophy (i.e., cerebral palsy) [36] and in patients with myelomeningocele [38].

### 5.3. Acquired Craniocerebral Disproportion

In newborn and infant patients, CSF shunting leads to early suture ossification with skull remodeling by formation of laminated layers of bone on the inner surface [36,67]. Approximation or overlapping of previously diastatic sutures and sinking of the fontanelle are predisposing factors [36,86]. A true secondary craniosynostosis with CCD occurs, with microcephaly and posterior fossa hypoplasia. As a result, impaired intracranial compliance with intracranial hypertension and cerebellar tonsillar herniation (acquired Chiari) may appear [21,68]. The latter may be facilitated by repeated lumbar punctures, external lumbar drainages or LPS that create a pressure gradient across the foramen magnum (Payner’s “pressure gradient hypothesis”) [69,87,88,89]. 

Shunt overdrainage will remain asymptomatic (silent period) while changes in ventricular size and CCD progress. Symptoms more often appear before the age of 10 years [83], with a peak between 6 to 9 years [60,90,91]. CCD is particularly frequent and severe in shunted patients with syndromic craniosynostosis or cranial bone abnormalities (osteopetrosis, achondroplasia, congenital hypoplasia of the posterior fossa) [70,92]. On the other hand, patients shunted after cerebral growth is complete are at a lower risk [89].

### 5.4. Venous Congestion Theories

According to the Monro-Kellie doctrine, a decrease in CSF volume and the induced negative ICP causes an increase in blood volume. This effect occurs mainly in the venous system, with a subsequent diffuse meningeal venous hyperemia and engorgement of the venous sinuses, which may be revealed on cranial and spinal MR images [51]. Increased elastance or stiffness appears, as this depends closely on the functioning of cerebral venous outflow mechanisms [26,93], making patients highly susceptible to even small changes in ICP [84]. Venous congestion may increase at the moment of shunt failure due to the increased ICP and a rapid collapse of large caliber bridging veins draining to the major venous sinuses [81]. This phenomenon is earlier and more severe in hydrocephalus of venous origin, congenital heart disease (increased pressure in the right cardiac chambers), pseudotumor cerebri, skull base abnormalities (achondroplasia, osteopetrosis, congenital hypoplasia of the posterior fossa, Chiari II) and syndromic craniosynostosis [26,94,95].

Recently, the “pulsatile vector theory” has been proposed and applied to the understanding of shunt overdrainage pathophysiology. Direct transmission of the CSF flow to the extrathecal compartment through the shunt produces a complete extinction of the centrifugal reflection wave at the aqueduct. The unchanged interstitial fluid shockwave leads to a ventricular collapse by progressive centripetal forces. In the subarachnoid space, the diastolic CSF pulse or “recurrent” wave (driving force of venous outflow at the cortical surface) may be reduced or eliminated and enlargement of venous vessels by increased volume load develops. Overdrainage symptoms appear whenever baseline ICP exceeds the venous pressure within the larger bridging veins, with increased outflow resistance causing venous congestion and cerebral edema. ICP may be further lowered by a resynchronization of physiological notch-filter and venous Starling-resistor, but the phenomenon would be irreversible with effect from a certain point [96].

Other capillary or venous mechanisms have been proposed. Jang described the so-called “capillary absorption laziness” theory. As CSF is not adequately absorbed in low or negative ICP conditions, in a situation of acute valve malfunction, the increased fluid pressures cannot be quickly compensated by a “lazy” or previously atrophied capillary system [30]. In the event of shunt malfunction, a rapid increase in ICP would lead to vein collapse and the interstitial fluid would not have enough time to dissipate through the venous system. Cerebral elastance increases and, as a result, ventricles fail to dilate and remain slit-like even in the face of a high ICP. On the other hand, according to Barami’s “cerebral venous overdrainage” theory, during CSF diversion the Starling resistor effect on draining cortical veins may be rendered nonfunctional, leading to overdrainage of cerebral venous blood during upright positioning through cranial venous outflow tracts. This might explain the venous engorgement in the cervical epidural veins causing radicular and myelopathic symptoms in patients with CSF loss. Cerebral venous overdrainage (CVO) could be the pathophysiological mechanism behind low pressure hydrocephalus, SVS and pseudotumor cerebri [97].

## 6. Management Strategies in Shunt Overdrainage

When the reduction in ventricular size occurs early (during the first months of life), the rate of symptomatic overdrainage is multiplied. This could indicate the need for primary (at the time of first implantation) or secondary (at the time of shunt revision) preventive action [98]. The strategy of valve exchange to prevent chronic overdrainage is well tolerated and seems to improve the clinical outcome in terms of ventricular width, symptom relief and revision rate. In some centers, protocols for shunted patients include the mandatory rule to counteract the hydrostatic pressure [99]. In hydrocephalic children, preventive strategies with adjustable differential pressure valves and integrated gravitational units have a good rate of shunt survival [100].

Current valve technology offers a great variety of options to reduce CFS flow across the valve: high-pressure valves, membrane ASD, flow-controlled devices (FCD), adjustable differential pressure valves, adjustable differential pressure valves combined with fixed gravitational units, adjustable gravitational valves (GV) or distal catheters having a smaller internal diameter [79,85,101,102,103,104].

All three main types of current ASDs (membrane ASD, FCD and GV) reduce the effect of siphoning but through a distinct mechanism. Not every device type is suitable for every patient [105]. In the case of membrane ASD, during verticalization, a switching diaphragm is progressively closed by the weight of the hanging hydrostatic column in the distal catheter. Additionally, the system works during inspiration in patients with a ventriculopleural shunt (inspiration may cause pressure gradients even when patients are lying down) [79,106]. These devices are ineffective with LPS [77]. Nevertheless, membrane ASD may stop drainage in the upright posture, leading to excessive accumulation. Besides, the membrane is highly susceptible to any kind of increased external tissue pressure, which can lead to functional obstructions [105]. Flow-controlled valves avoid increased drainage by closing their primary flow path when drainage exceeds a certain threshold. The second pathway is a longer and smaller diameter, resulting in increased flow resistance. This becomes relevant under high CSF differential pressure conditions [107,108]. However, these systems may limit rapid regulation in circumstances of high ICP [105]. On the other hand, no initial closing of the low resistance path is possible in an upright posture when the driving pressure gradient is not large enough (short patients who have a smaller hydrostatic pressure column) and re-opening has also been reported for slightly increased intraperitoneal pressure (obese patients with increased intraperitoneal pressure) [105]. FCD do not offer adjustability. Czosnyka found FCD to be helpful in overdrainage related to slow vasogenic ICP waves, as membrane ASD and GV might not be effective in such cases [77]. GV use small metallic balls as controlling weights instead of the hydrostatic column [102,108]. The Frankfurt horizontal plane should be used as a reference for implantation [109]. GV prevent uncontrolled drainage of the CSF during daytime activities, securing a minimum of recurrent CSF pulse wave to the subarachnoid space by a threshold amount of intraventricular CSF [96,110,111]. Independent opening pressures in horizontal and vertical orientation are allowed, but GV do not provide constant drainage in an upright posture. Differences in sitting and standing postures have been reported and precise adjustment to the height and intraperitoneal pressure is required. Consequently, adaptation to the individual patient is critical. In fact, non-programmable GV may not be advisable when the required opening pressure is difficult to estimate before surgery [105]. It should be taken into account that some patients with lower intracranial pressures may benefit from non-siphon control valves [112]. GV may cause under-drainage in bedridden patients [113].

In symptomatic patients, postural measures will help in transitory forms of intracranial hypotension (lumbar puncture, external lumbar or ventricular drainage or immediate postoperative CSF shunt) and in first episodes of presentation of postural headaches [84]. In refractory cases, cyproheptadine or topiramate have been employed but implantation of an in-line ASD could relieve symptoms both in children and adults in 85% of cases [114]. ASD and GV units may overcome the siphoning effect, but adjustability of GV has the advantage of higher treatment flexibility [115]. Some attempts have been made to determine the so called “vertical effective opening pressure” of the entire shunt system to determine the best adjustments for alleviating symptoms regarding the implantation of GV [116]. ICP measurements may be helpful in identifying the appropriate individual valve setting [115,117]. Changes in patient weight, height, mobility and activity should also be considered [118]. Adjustable GV may allow maintaining low pressure settings in decubitus in NPH patients, with significantly better outcomes and lower risk of subdural hematomas. In contrast, with differential pressure valves the pressure settings of the valve have to be rather high to prevent overdrainage, thereby potentially provoking underdrainage in a horizontal position [119]. Continuous clinical improvement may be obtained by decreasing opening pressure (if necessary, down to 0 mmH_2_O) and secondary responses can be obtained in initially non-responder patients [120,121]. NPH may even evolve to a “ultra-low-pressure” hydrocephalic state and downgrading valves and gravitational units may facilitate a “therapeutic siphoning” effect [122,123,124]. Barami’s theory of uncontrolled cerebral venous overdrainage might explain why patients remain symptomatic even despite the addition of an ASD [97].

Once the ventricular system has collapsed, correction of the siphon effect may not be enough to control symptoms of overdrainage. Repositioning of the shunt can be carried out from the collapsed ventricle to another dilated or isolated area of the ventricular system, even to a porencephalic cavity. Neuronavigation or neuroendoscopic assistance is recommended as septostomies or transeptal multiperforated catheters are sometimes necessary [82,84,125,126]. The repositioned shunt system should be optimized according to the characteristics of the new proximal catheter location: in porencephalic areas high resistance systems may not be necessary as collapse onto the catheter tip is unlikely to occur. Neuroendoscopic or microsurgical fenestration may be necessary to treat a trapped fourth ventricle [127,128,129].

When the ventricular system is totally collapsed, shunt transferal is both difficult and risky. A protocol of controlled dilatation (the so-called shunt removal protocol) may be necessary by externalization of the shunt with continuous ICP recording. In some cases, these schemes allow a shunt independence state to be achieved when ventricular dilatation is tolerated with no increase in the ICP. In cases with symptomatic ventricular dilatation or raised ICP, ETV may be considered [20,26,130], although a favorable etiology and anatomical MRI findings should be considered for patient selection. Some authors consider that patients with myelomeningocele should be excluded [26]. Secondary ETV at the time of shunt malfunction may well be successful whereas planned removal of the shunt in overdrainage is an invasive procedure with more likelihood of failure [131]. In cases of increased ICP without ventricular dilatation, the condition behaves as a pseudotumor cerebri (“shunt pseudotumor”) with CSF isolation in the subarachnoid space, and LPS implantation has been suggested. This procedure may be adequate for chronic overdrainage management in hydrocephalus of venous origin [26]. Drainage of the subarachnoid space permits a certain degree of ventricular dilatation by creating a favorable pressure gradient through the cortical mantle [23]. As the proximal catheter is located in the thecal sac, there is no possibility of collapse of this reservoir [132]. High-resistance valves or FCD should be considered to prevent overdrainage (including acquired Chiari development) [89,132,133,134]. Adjustable non-siphon-controlled valves have been successfully employed in LPS for patients with a diagnosis of idiopathic intracranial hypertension [135]. Ideally, these systems should harbor distal catheters having a smaller internal diameter. When patients are not candidates for LPS, drainage of CSF from the cisterna magna has been proposed [136].

In patients with CCD, shunt optimization, transferal, communication of isolated cavities, shunt removal or LPS may not be able to control symptoms of overdrainage. Particularly, patients with skull base abnormalities and a compromised foramen magnum (stenosis of the jugular foramen, Chiari I, myelomeningocele with Chiari II, achondroplasia, osteopetrosis or syndromic craniosynostosis) may have venous congestion associated with CCD. Although a short course of steroids helps control the symptoms [83] and antimigraine drugs may reduce the venous congestion [137], medical treatment is only justified while definitive therapy is planned. In these cases, cranial expansion procedures should be considered. Nowadays, suturectomies or subtemporal decompression techniques [13,14,138] have been abandoned and parietal and posterior cranial vault expansion [21,139,140,141] or internal cranial decompression techniques [142,143] are preferred. In patients with SVS and CCD, cranial expansion produced long-lasting relief of symptoms and a significant reduction in the mean ICP and ICP spikes [21]. Acquired Chiari may also be addressed by supratentorial cranial expansion [70] with the advantage that it can be used independently of whether the patient harbors a LPS or VPS and it does not require valve manipulation. LPS should be converted to VPS before suboccipital decompression [69,144,145]. On the other hand, patients with sinking skin/bone flap symptoms improve after correction of the overdrainage and cranioplasty.

## 7. Proposal for an Integrative Model in Shunt Overdrainage

Despite the superb contributions made by many authors on the topic of shunt overdrainage in the last half century, severe forms of shunt overdrainage syndrome still exist and an active prevention policy is lacking in many centers. Universally agreed-upon diagnostic criteria and classifications are necessary, probably based on a better understanding of the pathophysiological mechanisms and relationships among them. In this work we have proposed an integrative model of pathophysiology with clinical manifestations, radiological findings and ICP patterns to help determine the best (sometimes stepped) management strategy. Differences between pediatric patients and adults are based on the different etiologies, the characteristic CCD that occurs in infants and the duration of the overdrainage period and its influence on the complexity and severity of stablished pathophysiological mechanisms. Figure 4 offers an algorithm for management based on pathophysiology and Figure 5 illustrates the idea. The model is summarized in Table 1. Shunt optimization by prevention or correction of siphoning with GV or ASD is mandatory as the first therapeutic measure to avoid or minimize CSF isolation, CCD and venous hypertension, making shunt overdrainage less severe and easier to manage. Shunt optimization or transferal and neuroendoscopy may be recommended when ventricular collapse and cerebrospinal fluid isolation appear. Cranial expansion may be useful in congenital or acquired craniocerebral disproportion and shunting the subarachnoid space in communicating venous hydrocephalus and idiopathic intracranial hypertension.

## Figures and Tables

**Figure 1 jcm-10-03620-f001:**
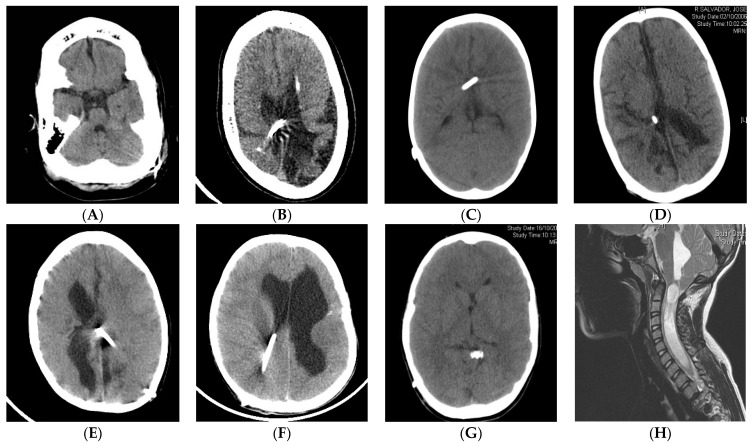
(**A**,**B**) A 5-year-old patient, shunted at birth after posthemorrhagic hydrocephalus. Postural symptoms. White matter damage with brain atrophy, small non-collapsed ventricles, extra-axial fluid collections, widened brain sulci, calcifications, laminated thickening of the calvarium, dolichocephaly and sinus hyperpneumatization. Shunt exchange with GV implantation was carried out, followed by clinical improvement. (**C**) An 8-year-old, shunted at birth after posthemorrhagic hydrocephalus. Complete ventricular collapse. Clinically, the patient presented sustained intracranial hypertension with papilledema, altered consciousness and bradycardia. Symptoms improved after shunt optimization by implanting an “in line” adjustable valve harboring a flow control device. (**D**–**F**) Ventricular collapse may be incomplete (**D**), unilateral (**E**) or focal (**F**). Non-collapsed areas may increase in size as a result of CSF isolation (**E**,**F**) and therefore can be considered for shunt transferring. (**G**) The catheter tip may be located outside the ventricular system (cisterns or extra-axial space). (**H**) Isolated fourth ventricle and syringomyelia as a result of sustained supratentorial overdrainage.

**Figure 2 jcm-10-03620-f002:**
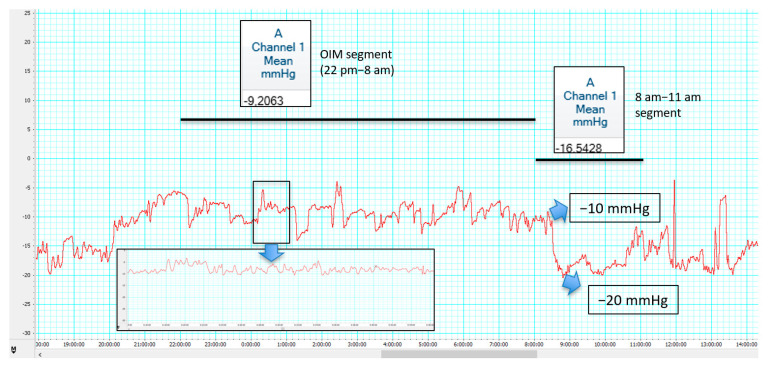
A 42-year-old female, pseudotumor cerebri treated by lumboperitoneal shunting (adjustable non-syphon control device). Chronic non-postural headache. Fundoscopy not fully reliable due to previous sequelae of papilledema. ICP monitoring revealed negative pressures during overnight ICP monitoring (OIM) segment (2200 to 0800) with a mean pressure of −9 mmHg, presenting a fall from −10 to −20 mmHg at the time of the tilting test after awakening, followed by a discrete rise with a mean of −16 mmHg in the interval from 0800 to 1100. Slow wave segments were also found (e.g., B waves up to 5 mmHg in amplitude from 2400 to 0100). Intracranial hypotension was diagnosed and valve upgrading obtained symptomatic improvement. Codman Microsensor^®^ ICP Transducer, Codman^®^ ICP EXPRESS Monitor^®^ (Integra LifeSciences Corporation, Princeton, NJ, USA)/ML856 PowerLab 26T, ML132 Bio Amp ADInstruments, INC/Personal Computer Chart v.8 software (ADInstruments, Dunedin, New Zealand).

**Figure 3 jcm-10-03620-f003:**
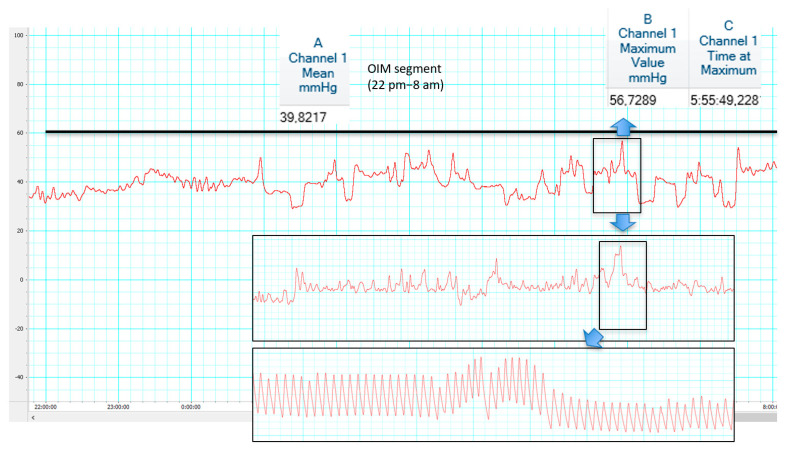
An 8-year-old patient with syndromic craniosynostosis and hydrocephalus, shunted at birth (also see Figure 5). Symptoms only partially improved after shunt optimization. ICP monitoring revealed irregular traces and raised ICP during OIM segment (2200 to 0800) with mean ICP of 39 mmHg and ICP peak over 56 mmHg at 05:55. High amplitude B waves (up to 20 mmHg) were found in slow wave segments where ICP pulse amplitude reached 20 mmHg and waveform had a triangular shape. Re-do posterior cranial vault gradual distraction was performed. Codman Microsensor^®^ ICP Transducer, Codman^®^ ICP EXPRESS Monitor^®^ (Integra LifeSciences Corporation, Princeton, NJ, USA)/ML856 PowerLab 26T, ML132 Bio Amp ADInstruments, INC/Personal Computer Chart v.8 software (ADInstruments, Dunedin, New Zealand).

**Figure 4 jcm-10-03620-f004:**
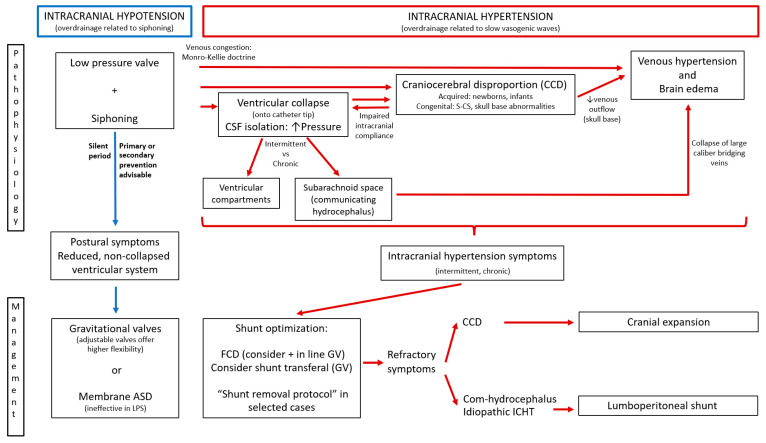
Shunt overdrainage syndrome. Integration of pathophysiological theories and algorithm for management. ASD: antisiphon device; FCD: flow control device; GV: gravitational valve; ICHT: intracranial hypertension.

**Figure 5 jcm-10-03620-f005:**
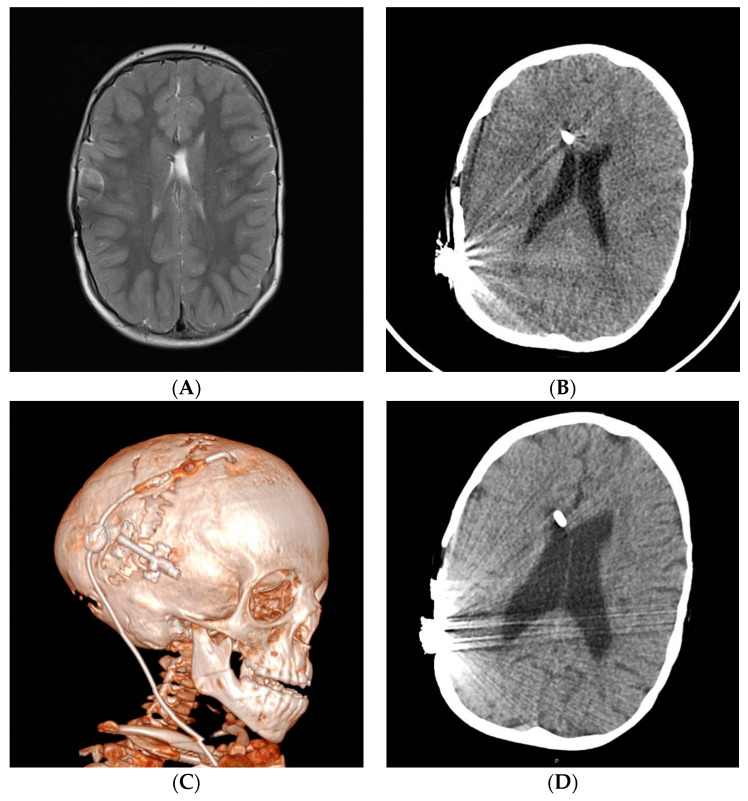
An 8-year-old patient with syndromic craniosynostosis and hydrocephalus, shunted at birth (also see Figure 3). Posterior cranial vault expansion by distraction osteogenesis performed before 1 year of age. The patient harbored an adjustable valve with FCD and in-line fixed antigravitational device. Episodes of headache and stiffness appeared, with slow valve refilling after pumping. (**A**) Ventricular collapse on MRI. (**B**) Improvement in ventricular size after shunt optimization (upgraded GV implanted in line) but symptoms only partially improved. (**C**) Re-do posterior cranial vault gradual distraction was performed with clinical and (**D**) significant radiological improvement.

**Table 1 jcm-10-03620-t001:** Integrative Model for SOS.

SOS Integrative Model	Intracranial Hypotension	Intracranial Hypertension		
Pathophysiology (Multifactorial)	Siphoning (Triggers Other Mechanisms)	Ventricular Collapse and CSF Isolation	Acquired CCD	Venous Hypertension
Description	Role of postural change (Law of Stevin). Venous congestion appears (Monro-Kellie). CVO when Starling resistor is non-functional	Ventricular collapse, distortion of cerebral structures with CSF isolation in ventricular compartments or in the subarachnoid space	Early suture ossification, microcephaly and acquired Chiari. Impaired cerebral compliance. Venous congestion is present	Increased pressure in the subarachnoid space, collapse of large bridging veins, venous hypertension, brain edema and increased cerebral elastance
Predisposing factors	Low brain elastance (brain atrophy). Infants: PHH, AS. Adults: previous SDH, SDHy, PTH, LOVA, NPH	Shunting at first months of life (PHH, neonatal meningitis), good brain elastance, low pressure valves	Early shunting (premature, newborn, infants), S-CS and skull base abnormalities	Hydrocephalus with venous origin, congenital heart disease, pseudotumor cerebri (idiopathic ICHT), S-CS and skull base abnormalities
Clinical manifestations	Chronic: variable silent period, “low-pressure postural headache”, etc. Sinking skin flap or OSAM may appear	Intermittent headaches (SVS): severe headache, nausea-vomiting, altered consciousness, etc. Chronic symptoms may appear between symptomatic crises.	Chronic states: weakness, irritability, poor food tolerance, developmental delay, etc. Acute episodes may appear	Symptoms and signs of sustained ICHT: papilledema, altered consciousness, bradycardia or systemic hypertension
Radiological findings	Small ventricles (moderate or normal sized also found). Extra-axial collections of fluid or blood. MRI: widened brain sulci, calcifications, etc.	Small ventricles with collapse onto the catheter tip. Occasionally with ventricular asymmetry, isolated compartments or IFV	Small ventricles. Suture sclerosis, laminated thickening, posterior fossa hypoplasia, etc. MRI: obliterated subarachnoid space, Chiari, spinal canal stenosis	Small ventricles with collapse onto the catheter tip. MRI: diffuse meningeal and arachnoid thickening, gadolinium hyperuptake, venous congestion, brain edema
ICP patterns	“Siphoning” pattern: Low/negative Pb Decrease < −10 mmHg in tilting tests Infusion tests: Pb < 0, Rcsf < 8, pulse amplitude < 4, RAP < 0.6	“Overdrainage related to ICP slow vasogenic waves”: Irregular traces, triangular shape of ICP pulse waveform, B waves storms Mean ICP > 20 mmHg, ICP peaks > 25 mmHg, RAP peaks > 0.6 Raised AMP and SLOW during the wave episodes Infusion tests: Pb usually high but pulse wave rarely visible		
Management strategies Primary or secondary prevention advisable	GV or Membrane ASD (ineffective in LPS)	1st. Shunt optimization: FCD recommended, consider combination with in-line GV. (Consider shunt transferal or “shunt removal protocol” in selected cases) 2nd. Cranial expansion in congenital or acquired CCD, particularly in S-CS and skull base abnormalities 3rd. LPS in communicating hydrocephalus or idiopathic ICHT. High resistance systems recommended		

Footer: AS: aqueductal stenosis; ASD: antisiphon device; CCD: craniocerebral disproportion; CVO: cerebral venous overdrainage; FCD: flow control device; GV: gravitational valve; ICHT: intracranial hypertension; ICP: intracranial pressure; IFV: isolated fourth ventricle; LOVA: long-standing overt ventriculomegaly in adults; LPS: lumboperitoneal shunt; MRI: magnetic resonance imaging; NPH: normal pressure hydrocephalus; OSAM: overshunting associated myelopathy; Pb: baseline pressure; PHH: posthemorrhagic hydrocephalus; PTH: post-traumatic hydrocephalus; RAP: regression of amplitude and pressure index; Rcsf: CSF outflow resistance; S-CS: syndromic craniosynostosis; SDH; subdural hematoma; SDHy; subdural hygroma; SOS: shunt overdrainage syndrome; SVS: slit ventricle syndrome.

## Data Availability

No new data were created or analyzed in this study. Data sharing is not applicable to this article.

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
