# Peer review of "Shunt Overdrainage: Reappraisal of the Syndrome and Proposal for an Integrative Model"

_jcm, 2021, doi:10.3390/jcm10163620_

Round 1
Reviewer 1 Report
A great deal of thought and energy is shown by this review of problems related to the functioning of shunts in shunt dependent patients. Its goal is to provide a new “integrated” model related to overdrainage. By integrated the authors meant that it would incorporate all aspects of hydrocephalus and shunting specific to the individual. They recommend prophylactic measures such as devices to prevent siphoning is mandatory. Since the abstract is what most readers will see and decide whether or not to go to the next level it would seem that the model should be shown in a concise manner.
In the introduction there is an extensive discussion of the history of hydrocephalus and its management. This was interesting but not important to the message of the review. The discussion of the negative aspects of shunt overdrainage (probably beginning with Portnoy) is consistent with the message. Most of 1: introduction is not essential to the discussion.
Relative to the “model,” Numbers 2-5 represent a “check sheet” in that they discuss the clinical syndromes, radiographic, ICP and physiological aspects related to an individual. In deciding on a clinical management strategy (#6) it would be necessary to have included the above aspects. I suggest that Tables for numbers 2-5 would help in making the point that what is sought here is an “integrated” model.
I would suggest that figures 2 and 3 are not useful and should be discarded. I am very confused by Figure 4. It would be good to have an algorithm for management that includes the results of 2-5 but this is not helpful.
I also think that 4.ICP patterns should be left out. This discussion is among engineering type is specific and limited programs. I suggest removing 4 altogether.
The discussion of pathophysiological theories includes complicated concepts that are outside of the important clinical patterns. It is clear that high venous pressures in the dural venous sinuses do play a role primarily in patients shunted in early infancy. However, the role of the veins in other forms of hydrocephalus are conjecture.
The table showing the entire Integrative model is too complex. It does not need the references in it since they are in the text. The nomenclature in this table should be the same as that used in the text.
Author Response
Point 1: Since the abstract is what most readers will see and decide whether or not to go to the next level it would seem that the model should be shown in a concise manner.
Response 1: Abstract has been re-written to show the proposed model in a concise manner. Particularly, recommendations for siphoning prevention and individualized management of shunt overdrainage based on specific pathophysiological mechanisms have been underlined
Point 2: In the introduction there is an extensive discussion of the history of hydrocephalus and its management. This was interesting but not important to the message of the review. The discussion of the negative aspects of shunt overdrainage (probably beginning with Portnoy) is consistent with the message. Most of 1: introduction is not essential to the discussion.
Response 2: Introduction has been shortened by cutting some historical information
Point 3: Relative to the “model,” Numbers 2-5 represent a “check sheet” in that they discuss the clinical syndromes, radiographic, ICP and physiological aspects related to an individual. In deciding on a clinical management strategy (#6) it would be necessary to have included the above aspects. I suggest that Tables for numbers 2-5 would help in making the point that what is sought here is an “integrated” model.
Response 3: Thank you for your comments. We preferred to consider treatment concepts (“6. Clinical management strategy”) separately to finally propose the integration with aspects described in numbers 2-5 under the subheading “7. Summary. Proposal for an integrative model in SOS” and in Table 1. A complete Table was chosen instead of different minor tables where the integration of concepts was no easy to visualize.
Point 4: I would suggest that figures 2 and 3 are not useful and should be discarded. I am very confused by Figure 4. It would be good to have an algorithm for management that includes the results of 2-5 but this is not helpful.
Response 4: From our point of view, Figures 2 and 3 are necessary not only to illustrate “4. ICP patterns” but also to explain how the integrative model may help in the management of complex cases. Figure 4 has been modified to offer an algorithm for management
Point 5: I also think that 4.ICP patterns should be left out. This discussion is among engineering type is specific and limited programs. I suggest removing 4 altogether.
Response 5: From our point of view, ICP monitoring should be considered a useful tool in shunt overdrainage management, especially in selected cases. In these patients, the identification of the ICP pattern is necessary and Figures 2 and 3 are an example of this. Although different hardwares and softwares have been developed, basic common concepts as mean ICP, tilting test or slow waves may help us to classify the type of overdrainage and to design tailored management strategies
Point 6: The discussion of pathophysiological theories includes complicated concepts that are outside of the important clinical patterns. It is clear that high venous pressures in the dural venous sinuses do play a role primarily in patients shunted in early infancy. However, the role of the veins in other forms of hydrocephalus are conjecture.
Response 6: From our point of view, failure to obtain clinical improvement in complex cases of shunt overdrainage is partially explained by the lack of consideration on the role of veins. Venous involvement has been reported in the so-called venous hydrocephalus, congenital heart disease, pseudotumor cerebri and skull base abnormalities. Although some exposed ideas can be considered nowadays only as “theories”, physicians involved in the management of these patients should know these plausible mechanisms.
Point 7: The table showing the entire Integrative model is too complex. It does not need the references in it since they are in the text. The nomenclature in this table should be the same as that used in the text
Response 7: References in Table 1 have been eliminated and nomenclature revised.
Reviewer 2 Report
This is a very well written and comprehensive overview describing the issue of shunt overdrainage and cites a multitude of previous studies performed to date describing this issue. I recommend publication with minor revisions after certain comments are met. The article is well written.
Comments:
The authors propose treatment steps at the end of this manuscript in the table and in the paragraph, is there any way to create a flow sheet figure to allow easy visualization to readers of what treatments steps should be taken?
Can the authors make a comment on whether this treatment proposal should be undertaken in both adult and pediatrics or if it differs among the two groups?
Can you further clarify what sequenced strategies are in line 95? Im not sure if this would be immediately evident to the reader
Minor Editorial Comments:
Remove probably throughout the manuscript.
Remove "making this condition less severe and easier to manage" line 26-27
Consider making the introduction subheading one phrase only and not three separate phrases.
I would modify line 92 and phrase it more along the lines of "several grading systems or proposed classification systems" exist
Remove abreviations from subheaders
Remove "so" line 260
Author Response
Point 1: The authors propose treatment steps at the end of this manuscript in the table and in the paragraph, is there any way to create a flow sheet figure to allow easy visualization to readers of what treatments steps should be taken?
Response 1: Figure 4 has been re-elaborated to offer an algorithm for management
Point 2: Can the authors make a comment on whether this treatment proposal should be undertaken in both adult and pediatrics or if it differs among the two groups?
Response 2: The comment has been included in 7. Summary
Point 3: Can you further clarify what sequenced strategies are in line 95? Im not sure if this would be immediately evident to the reader
Response 3: Done
Point 4: Minor Editorial Comments:
Response 4: Done
Round 2
Reviewer 1 Report
the authors have taken the revision of the manuscript seriously and no further revisions are needed